# Structure of the activated Edc1-Dcp1-Dcp2-Edc3 mRNA decapping complex with substrate analog poised for catalysis

Jeffrey S. Mugridge[1], Ryan W. Tibble[1,2], Marcin Ziemniak[3,4,5], Jacek Jemielity[4] & John D. Gross [1,2]

The conserved decapping enzyme Dcp2 recognizes and removes the 5′ eukaryotic cap from mRNA transcripts in a critical step of many cellular RNA decay pathways. Dcp2 is a dynamic enzyme that functions in concert with the essential activator Dcp1 and a diverse set of coactivators to selectively and efficiently decap target mRNAs in the cell. Here we present a 2.84 Å crystal structure of *K. lactis* Dcp1–Dcp2 in complex with coactivators Edc1 and Edc3, and with substrate analog bound to the Dcp2 active site. Our structure shows how Dcp2 recognizes cap substrate in the catalytically active conformation of the enzyme, and how coactivator Edc1 forms a three-way interface that bridges the domains of Dcp2 to consolidate the active conformation. Kinetic data reveal Dcp2 has selectivity for the first transcribed nucleotide during the catalytic step. The heterotetrameric Edc1–Dcp1–Dcp2–Edc3 structure shows how coactivators Edc1 and Edc3 can act simultaneously to activate decapping catalysis.

---

[1] Department of Pharmaceutical Chemistry, University of California, San Francisco, San Francisco, CA 94158, USA. [2] Program in Chemistry and Chemical Biology, University of California, San Francisco, San Francisco, CA 94158, USA. [3] Division of Biophysics, Institute of Experimental Physics, Faculty of Physics, University of Warsaw, 02-089 Warsaw, Poland. [4] Centre of New Technologies, University of Warsaw, 02-097 Warsaw, Poland. [5]Present address: Biological and Chemical Research Centre, Department of Chemistry, University of Warsaw, 02-089 Warsaw, Poland. Correspondence and requests for materials should be addressed to J.D.G. (email: jdgross@cgl.ucsf.edu)

Degradation of mRNA transcripts plays a critical role in maintenance of the cellular transcriptome and in control of eukaryotic gene expression[1]. Removal of the m7GpppN cap structure found at the 5′ end of all eukaryotic mRNA marks these transcripts for rapid clearance via the 5′-to-3′ mRNA decay pathway[2]. The conserved mRNA-decapping enzyme Dcp2 catalyzes hydrolysis of the 5′ cap to liberate m7GDP and 5′ monophosphate RNA, which can then be degraded by conserved 5′-3′ exonucleases[3–6]. Dcp2-mediated decapping is a critical step in many cellular RNA decay pathways including bulk 5′–3′ decay[4,5], nonsense-mediated decay[7–10], microRNA-mediated decay[11,12], decay of long-noncoding RNA[13], and decay of transcripts containing non-optimal codons[14]. Recently, Dcp2 was also shown to be a reader of reversible m6A modifications in the 5′ cap, whereby m6A methylation inhibits decapping and stabilizes cellular mRNA transcripts[15].

Dcp2 consists of an N-terminal regulatory domain (NRD) and a catalytic Nudix domain (CD)[16,17] linked by a short, flexible hinge that allows the enzyme to dynamically adopt different conformations in solution[18,19] and in crystal structures[20–24]. The CD harbors a positively charged patch for RNA binding (box B motif)[25,26] and a Nudix motif with conserved Glu residues that bind magnesium and carry out cap hydrolysis chemistry[27,28]. Fungal Dcp2 also contains an unstructured C-terminal extension that harbors protein–protein interaction motifs and elements that inhibit Dcp2 catalysis[29]. At least some of these motifs are transferred to the C-terminus of the essential activator Dcp1 in metazoans, suggesting a conserved biological function[30]. The NRD enhances decapping activity by specifically recognizing the m7G nucleotide of cap and binding Dcp1, which further accelerates decapping[17,19,21,22,31,32].

The activity of the core Dcp1–Dcp2 complex can be further modulated by the enhancer of decapping proteins (Edcs)[33]. For example, the Edc1-type activators, which include the yeast paralog Edc2 and PNRC2 in metazoans, contain conserved tandem short linear motifs to bind Dcp1 and activate Dcp2 catalysis[19–21,34–37]. Edc3 activates decapping by promoting RNA binding and alleviating autoinhibition of Dcp2 in yeast[22,29,38–42]. Edc3 is recruited to the decapping complex via short linear motifs found in the fungal or metazoan Dcp2 or Dcp1 C-terminal extension, respectively[29,33]. Edc4 is a metazoan-specific scaffolding protein that strengthens Dcp1–Dcp2 interaction and coordinates decapping with exonucleolytic degradation by Xrn1[32,43–45]. Genetic and physical interactions between activator proteins suggest these cofactors work together to promote decapping, but how this is achieved at the structural level is unknown[45–49].

Several recent structural studies have provided new insights into how Dcp2 recognizes the m7G cap, how Dcp2 activity is enhanced by coactivators, and how Dcp2 solution conformations are tied to catalysis. First, a study by Valkov et al. presented a structure of *S. pombe* Edc1–Dcp1–Dcp2 in which the activation motif in Edc1 bridges the Dcp2 NRD and CD to apparently promote a conformation of Dcp2 that enhances RNA binding[20]. Soon after, our lab published the first substrate analog-bound structure of *S. pombe* Dcp1–Dcp2 in complex with human PNRC2[21]. Our structure showed a very different conformation than the Valkov structure, in which the substrate analog was bound by a composite binding site using conserved residues on the NRD and CD. However, in this pre-catalytic conformation of Dcp2 the catalytic Glu on the Nudix helix was positioned too far from substrate for effective catalysis, and the activation motif of PNRC2 was disordered, suggesting an additional conformational change in Dcp2 was needed to achieve catalysis. Simultaneously, Charenton et al. published a *K. lactis* Dcp1–Dcp2–Edc3 structure with bound m7GDP product in which m7G was recognized by the

Dcp2 NRD as in our Mugridge et al. substrate-bound structure, but the CD was rotated by ~90° to bring the Nudix helix much closer to the cleaved m7GDP product, consistent with an active conformation of Dcp2[22]. Finally, most recently Wurm et al. published a study that presented an *S. pombe* Edc1–Dcp1–Dcp2 product-bound structure in the same conformation as observed by Charenton et al., a conformation that is significantly populated in solution when Edc1 and m7GDP or capped RNA are bound[19].

In this study, we present a new, four-protein structure of *K. lactis* Edc1–Dcp1–Dcp2–Edc3 at 2.84 Å resolution in the catalytically active conformation with bound substrate analog poised for catalysis. We show that the conserved YAGxxF activation motif of Edc1 mediates a three-way interface between Dcp1 and the Dcp2 NRD and CD, and that lesions on the NRD and in the flexible hinge linking the CD and NRD abolish activation by Edc1. Substrate analog binding to the active conformation of Dcp2 suggests that the first transcribed nucleotide of RNA is recognized by base stacking to a conserved aromatic residue at the top of the RNA-binding channel, and that the RNA body follows this positively charged path to bound Edc3 coactivator. We show that Dcp2 has selectivity for the first transcribed nucleotide of RNA substrate and that this selectivity depends on the conserved aromatic residue that binds this nucleotide of our substrate analog. Overall, these data show how substrate binds and is recognized by the catalytically active conformation of Dcp2, suggest a path for RNA binding consistent with prior in vitro mutagenesis and binding data, and show how coactivators Edc1 and Edc3 can simultaneously bind Dcp2 to enhance mRNA decapping by the Dcp1–Dcp2 complex. Our structure establishes that the active conformation of Dcp2 is the same for both substrate and product-bound complexes, and clarifies the mechanisms of decapping catalysis and control of decapping activation by multiple protein interactions.

## Results

**Structure of substrate analog-bound Edc1–Dcp1–Dcp2–Edc3.** To obtain our recently published substrate analog-bound PNRC2–Dcp1–Dcp2 structure[21], we crystallized an apo PNRC2–Dcp1–Dcp2 complex and soaked in cap analog to induce a conformational change and substrate binding within the crystal. While this substrate-bound conformation gave new insight into cap recognition by Dcp2, it was clear that this did not represent the catalytically active conformation because the catalytic Glu residue on the Nudix helix was positioned to too far from bound substrate to carry out hydrolysis chemistry. When we attempted to push Dcp2 into the active conformation by soaking in catalytically required magnesium along with substrate analog, the crystals were physically damaged and diffraction was completely destroyed, suggesting this induced an additional conformational change in Dcp2 that could not be accommodated by the crystal lattice.

To trap the substrate-bound active conformation, we screened different Dcp1–Dcp2 constructs (*S. pombe*, *K. lactis*, *H. sapiens*) in which the catalytic Glu was mutated to Gln to prevent cap hydrolysis for co-crystallization with magnesium, a tight-binding two-headed cap analog[50], and coactivators Edc1 and Edc3 (Fig. 1a,b). The *K. lactis* constructs crystallized most readily and we obtained a structure of *Kl* Edc1–Dcp1–Dcp2–Edc3 with bound magnesium and two-headed cap analog at 2.84 Å resolution (Fig. 1c, Table 1, solved by molecular replacement with PDB 5LOP[22] Dcp1–Dcp2). Dcp2 is in the catalytically active conformation, very similar to that found in recently reported m7GDP product-bound structures (Supplementary Fig. 1; Dcp2 core backbone RMSD is 1.2 Å with PDB 5LOP and 1.9 Å with PDB 5N2V)[19,22], with the Nudix helix positioned close to bound

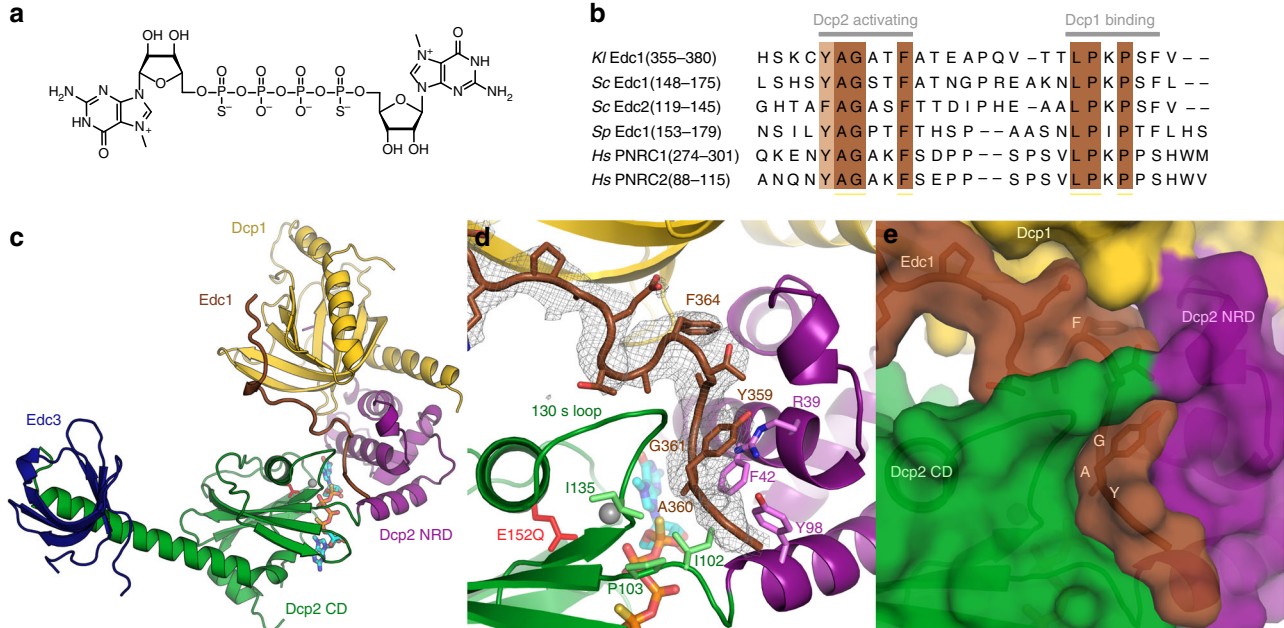

**Fig. 1** Structure of the activated decapping complex poised for catalysis. **a** Structure of the high-affinity, two-headed cap substrate analog. **b** Sequence alignment of Edc1-like coactivator peptides showing the conserved Dcp2-activating and Dcp1-binding motifs; background color intensity is scaled to conservation. **c** Structure of the substrate analog-bound *Kl* Edc1–Dcp1–Dcp2–Edc3 complex. Edc1 is brown, Dcp1 is yellow, Dcp2 NRD is purple, Dcp2 CD is green, Edc3 Lsm domain is dark blue, substrate analog is cyan. **d** Close-up view of the three-way interface between Dcp1, Dcp2 NRD, and Dcp2 CD mediated by the Edc1 YAGxxF Dcp2-activating motif. $F_o$-$F_c$ omit map for the Edc1 peptide is shown at 2.5$\sigma$. Sidechains for the Edc1 peptide and nearby conserved residues on Dcp2 are shown as sticks. The catalytic Glu on the Nudix helix, mutated in our structure to Gln, is labeled as E152Q in red. **e** Surface view of the Edc1 activation motif–Dcp1–Dcp2 interface, with conserved residues in the activation motif labeled

cap. Edc3 Lsm domain binds an extended helix at the C-terminus of the Dcp2 CD, as in the recent *Kl* Dcp1–Dcp2–Edc3 structure[22], and the fully ordered Edc1 activation peptide bridges Dcp1 and Dcp2, similar to the recent *Sp* Edc1–Dcp1–Dcp2 structure[19]. The co-occupancy of Edc1 and Edc3 coactivators in our structure highlights the idea that these decapping factors work together to promote decapping by complementary mechanisms[42].

**Edc1 consolidates the active conformation of Dcp2.** Edc1-like coactivator proteins contain tandem, conserved short linear motifs for Dcp1 binding and Dcp2 activation (Fig. 1b)[20,21,35,37]. Several structures have now been published that show how the proline-rich Dcp1-binding motif in Edc1-like peptides binds a conserved aromatic cleft on Dcp1[19–21,36], and our structure displays a very similar interaction between *Kl* Edc1 and *Kl* Dcp1 as that observed in *Sp*, *Hs*, and chimeric *Sp*–*Hs* structures (Supplementary Fig. 2). The conserved YAGxxF Dcp2-activating motif creates a 3-way interface that bridges Dcp1, Dcp2 NRD, and Dcp2 CD in the catalytically active conformation by threading through a narrow channel between these domains (Fig. 1d,e). In total, the Edc1 peptide interaction with the Dcp1–Dcp2 complex buries ~1300 Å$^2$ of surface area, with the YAGxxF Dcp2-acting motif accounting for ~500 Å$^2$ of this buried surface. The interface between the Dcp2 CD and Edc1 peptide covers only ~250 Å$^2$ of surface area, consistent with evidence that the peptide alone is insufficient to drive a conformational change in solution[19].

We have shown that Edc1-like proteins accelerate decapping catalysis by both increasing $k_{cat}$ and decreasing $K_M$.[21,35] Kinetic analysis of the Dcp2-activating YAGxxF motif in PNRC2 showed that mutation of the Tyr residue completely abolishes $k_{cat}$ activation of Dcp2 catalysis, while mutation of Ala and Phe residues significantly impairs activation[21]. We can now rationalize our previous biochemical observations in the context of the

fully activated, substrate analog-bound structure. Tyr359 of *Kl* Edc1, the residue most critical for activation, makes a cation–π interaction with the strictly conserved Arg39 on the Dcp2 NRD (Fig. 1d). Ala360 packs against conserved hydrophobic residues in the flexible hinge between Dcp2 NRD and CD, and Phe364 plugs into a pocket at the Dcp1–Dcp2 NRD interface consisting of positively charged and hydrophobic residues (Supplementary Fig. 3). These contacts anchor the Edc1 activation motif to Dcp1–Dcp2 and provide a complementary surface for the flexible hinge and 130s loop on the Dcp2 CD to pack against Edc1 in the catalytically active conformation of the enzyme. In this way, protein interactions mediated by the Dcp2 activation motif of Edc1 accelerate $k_{cat}$ not by contacting cap, but instead by limiting torsional degrees of freedom in the hinge to favor the active conformation of Dcp2, in which conserved residues on the CD and NRD recognize and hydrolyze cap using the composite active site.

**Residues in Dcp2 NRD and hinge are critical for activation.** In order to validate the Edc1–Dcp2 interface observed in our structure and better understand how the Edc1 peptide interacts with Dcp2 to promote catalysis, we mutated conserved residues on the Dcp2 NRD, CD, and hinge (Dcp2 residues shown as sticks in Fig. 1d) that contact YAG residues of the Edc1 peptide, and asked how these mutations affect activation of Dcp2 catalysis by Edc1. Using single-turnover in vitro kinetic decapping assays with cap-radiolabeled RNA substrate and *S. pombe* Edc1–Dcp1–Dcp2 complex, we find that residues Arg33 (*Kl* Arg39) and Ile96 (*Kl* Ile102) are absolutely essential for Edc1 function and that mutation of these conserved residues to Ala or Gly, respectively, completely blocks activation by Edc1 (Fig. 2). In our structure, Arg33 (*Kl* Arg39) makes a cation–π interaction with Edc1 Tyr359 to anchor the activation motif to Dcp2 NRD, and Ile96 (*Kl* Ile102)

**Table 1 Crystallographic data collection and refinement statistics**

|  | *Kl* Edc1–Dcp1–Dcp2–Edc3 with substrate analog (PDB 6AMO) |
|---|---|
| Data collection |  |
| Space group | C2 |
| Cell dimensions |  |
| $a$, $b$, $c$ (Å) | 174.26, 83.25, 104.30 |
| $\alpha$, $\beta$, $\gamma$ (°) | 90.00, 93.62, 90.00 |
| Resolution (Å) | 46.0–2.84 (3.00–2.84)[a] |
| $R_{merge}$ | 0.058 (0.78) |
| $I/\sigma I$ | 9.8 (1.3) |
| $CC_{1/2}$ | 99.8 (58.5) |
| Completeness (%) | 98.7 (99.1) |
| Redundancy | 2.9 (2.8) |
| Refinement |  |
| Resolution (Å) | 46.0–2.84 |
| No. reflections | 35,286 |
| $R_{work}/R_{free}$ | 0.2293 / 0.2491 |
| No. atoms |  |
| Protein | 8610 |
| Ligand | 115[b] |
| Water | – |
| B factors |  |
| Protein | 104 |
| Ligand | 114[b] |
| Water | – |
| R.m.s. deviations |  |
| Bond lengths (Å) | 0.003 |
| Bond angles (°) | 0.665 |
| Ramachandran plot statistics |  |
| No. favored | 967 (95.6 %) |
| No. allowed | 45 (4.4 %) |
| No. outliers | 0 (0 %) |

Data set was collected from a single crystal
[a] Values in parentheses are for highest-resolution shell
[b] Two molecules of two-headed cap analog and one magnesium ion

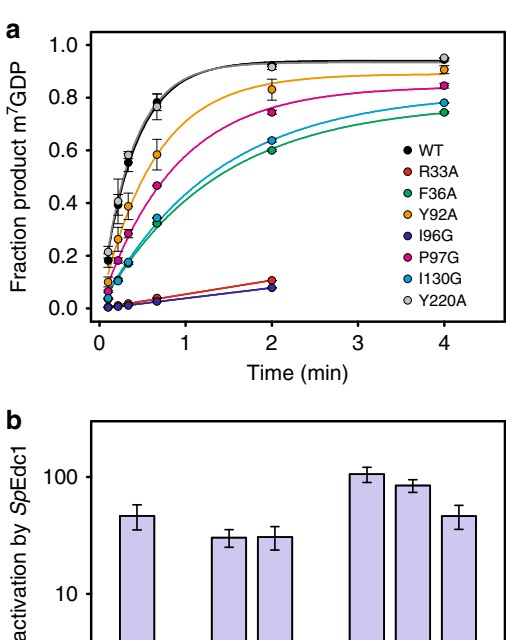

**Fig. 2** Mutations on both the Dcp2 NRD and flexible hinge prevent activation by Edc1. **a** In vitro decapping timecourses for *Sp* Edc1–Dcp1–Dcp2 WT and Dcp2 mutant complexes. R33A and I96G mutations completely block activation of Dcp1–Dcp2 by Edc1 peptide. Concentration of Dcp1–Dcp2(1–243) is 2 μM and Edc1(155–180) peptide is 100 μM. Errors are s.d. of two independent replicates. **b** Fold activation of WT and Dcp2 mutant decapping complexes by Edc1 peptide. Calculated as $k_{obs}$(Edc1–Dcp1–Dcp2)/$k_{obs}$(Dcp1–Dcp2). $k_{obs}$ values for Dcp1–Dcp2 with no Edc1 coactivator peptide are shown in Supplementary Fig. 4. Errors are s. d. of $k_{obs}$(Edc1–Dcp1–Dcp2)/$k_{obs}$(Dcp1–Dcp2) values obtained from two independent replicates

packs against Edc1 Ala360 and the Tyr359 backbone. These data suggest that both positioning of the peptide activation motif along the Dcp1–Dcp2 NRD surface, and specific van der Waals interactions with the Dcp2 hinge that links the NRD and CD, are critically important for Edc1-mediated activation.

Interestingly, mutation of other strictly conserved residues on Dcp2 that contact the Edc1 peptide in the catalytically active conformation, such as Phe36 (*Kl* Phe42) and Tyr92 (*Kl* Tyr98), have very little effect on Edc1's ability to activate Dcp2. Mutation of Tyr220, a distal residue on the CD that is important for conformational dynamics in Dcp2[42], also has no effect on activation by Edc1. Importantly, none of the tested mutations to conserved Dcp2 residues resulted in large defects in decapping in the absence of Edc1 (Supplementary Fig. 4), showing that Arg33 (*Kl* Arg39) and Ile96 (*Kl* Ile102) are critical specifically for Edc1-mediated activation of Dcp2. Finally, in contrast to our previous observations examining Dcp2 activation in the chimeric *Hs* PNRC2–*Sp* Dcp1–Dcp2 system where we showed PNRC2 peptides with the YAGxxF Dcp2-activating motif had little impact on the $K_M$ of the decapping complex, with native *Sp* Edc1–Dcp1–Dcp2 complexes we see a ~100-fold decrease in $K_M$ along with a ~10-fold increase in $k_{max}$ upon addition of the activator peptide (Supplementary Fig. 5; Supplementary Table 1). This is consistent with recent evidence that *Sp* Edc1 peptides can enhance RNA binding to the decapping complex[19,20].

**Substrate analog binds a composite active site on Dcp2.** Two-headed cap substrate analog bound at the Dcp2 active site engages conserved residues on the Dcp2 NRD and CD (Fig. 3a,b). In agreement with our previously reported substrate analog-bound pre-catalytic structure[21], and recently reported m⁷GDP product-bound structures[19,22], the m⁷G of cap is recognized by conserved, catalytically critical Trp and Asp residues on the Dcp2 NRD. The opposite face of this m⁷G nucleotide packs against the backbone of the Dcp2 CD 190s loop in the catalytically active conformation. This agrees with recent product-bound structures, but differs from the pre-catalytic conformation, in which this face of m⁷G was engaged in base stacking with *Sp* Tyr220 (*Kl* Phe223). In the active conformation reported here however, the CD is rotated ~90° from the pre-catalytic conformation, and Phe223 base stacks with the second m⁷G of our two-headed substrate analog, likely mimicking binding of the first transcribed nucleotide of RNA substrate to the Dcp2 CD. Additional conserved and catalytically important residues make contacts with the substrate analog phosphate chain, including the general acid Lys134 and Arg132 on the CD, and Lys99 on the NRD.

**Substrate analog positioning suggests a path for RNA binding.** Our structure suggests that the first transcribed nucleotide of RNA binds Phe223 at the top of an extended, positively charged

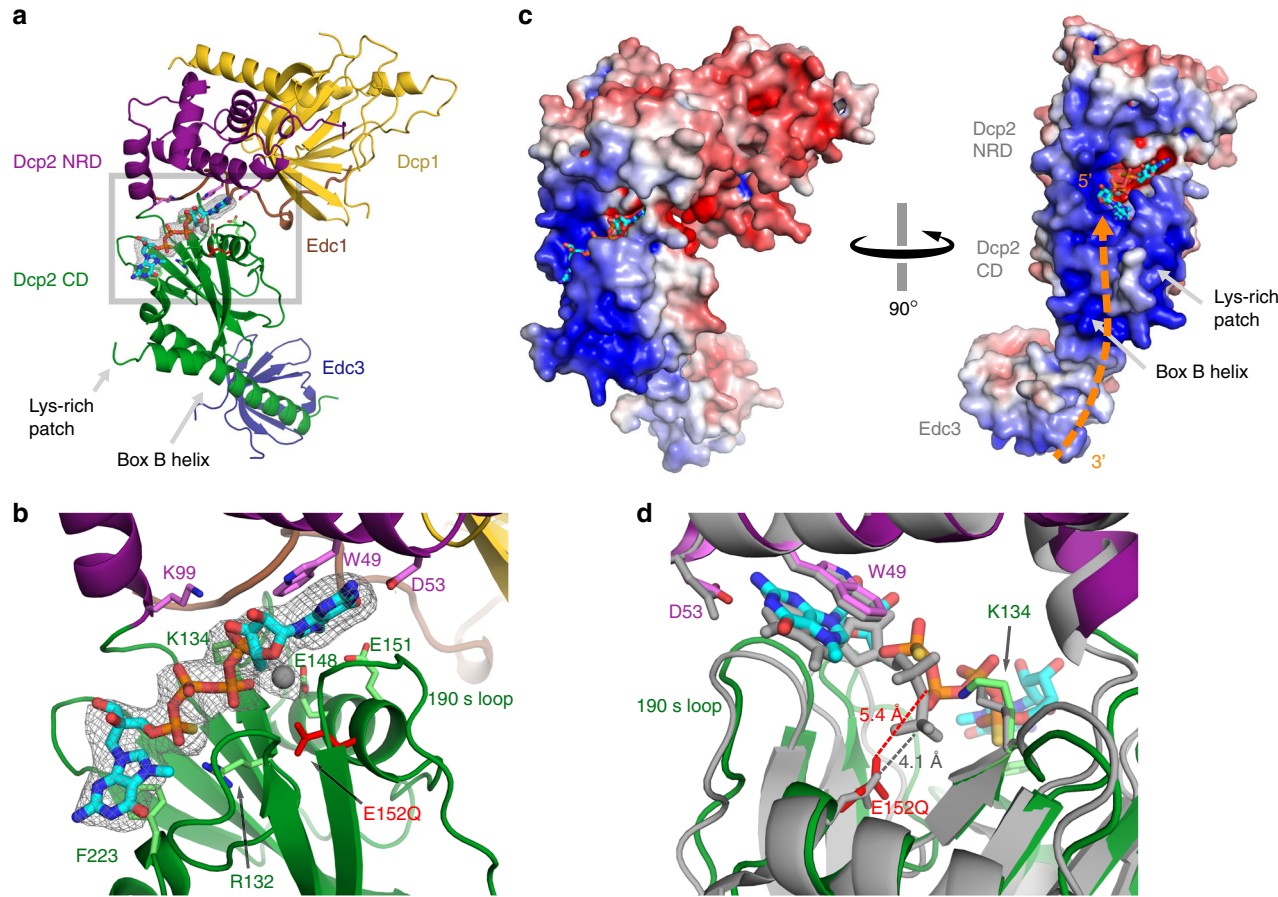

**Fig. 3** Substrate analog binding to the active conformation of Edc1–Dcp1–Dcp2–Edc3 decapping complex. **a** Full Edc1–Dcp1–Dcp2–Edc3 structure showing substrate analog binding between the Dcp2 NRD and CD. **b** Close-up of substrate analog bound at the Dcp2 active site. Conserved, catalytically important residues that contact the cap analog are shown as sticks. The catalytic Glu base (mutated to Gln in our structure) is colored red; $Mg^{2+}$ ion is shown as a gray sphere; $F_o$–$F_c$ omit map for cap analog and $Mg^{2+}$ is shown at $2.5\sigma$ in gray. **c** Edc1–Dcp1–Dcp2–Edc3 complex with calculated electrostatic surface (−6 to +6 kT/e) in the same orientation as in (**a**) (left), and rotated 90° (right). Dotted orange line shows the predicted path of RNA based on the orientation of substrate analog in the active site of Dcp2. The 5′ cap of RNA binds in the Dcp2 composite active site (substrate analog in cyan), RNA follows the positively charged Dcp2 CD C-terminal Box B helix past a conserved Lys-rich patch, to Edc3 coactivator. **d** Alignment of the substrate analog-bound (colored structure, PDB 6AM0) and product-bound (gray, PDB 5LOP [22]) *K. lactis* Dcp2 structures; view is rotated 180° from that shown in (**a**) and (**b**). The Dcp2 NRD (residues 8–97) was used for alignment, giving a backbone RMSD of 0.78 Å; all-atom RMSD for alignment of both the NRD and CD (residues 8–240) of substrate analog versus product-bound Dcp2 is 1.6 Å. Distance from catalytic Glu residue to β-phosphate of substrate analog or product are shown as red and gray dotted lines, respectively

C-terminal helix on the CD of Dcp2 (Box B helix, Fig. 3a). Based on this cap orientation, the RNA body likely extends down the positively charged Box B helix, past a conserved Lys-rich patch on the dorsal surface of Dcp2, to coactivator Edc3. This predicted RNA-binding path is consistent with many lines of biophysical and biochemical evidence, including: (1) NMR chemical shift perturbations induced by RNA binding on the surface of the Dcp2 CD[19,26], (2) biochemical data showing that the Box B helix is critical for decapping activity in vitro[25], (3) kinetic analysis showing that mutations to basic residues in both the Box B helix and Lys-rich patch significantly increase substrate $K_M$[26], (4) SPR data showing that mutations in the Box B helix significantly weaken RNA binding[17], (5) yeast growth assays showing that mutations in the Box B helix and Lys-rich patch result in temperature-sensitive growth defects[26], and (6) evidence that Edc3 activates decapping in part by enhancing RNA binding to the decapping complex[22]. The positioning of substrate analog in the Dcp2 active site revealed by our Edc1–Dcp1–Dcp2–Edc3 structure suggests an RNA-binding path along the Dcp2 Box B helix that nicely agrees with all of this previous RNA binding and activity data, adding structural

support to the RNA-binding path that has been hypothesized for some time by ourselves and others.

**Substrate and product-bound Dcp2 conformations are similar.** Alignment of the Dcp2 catalytic core from the active conformation substrate analog and product-bound structures shows that these adopt overall very similar Dcp2 domain orientations (Fig. 3d; Supplementary Fig. 1a). This is consistent with NMR chemical shift data that suggest the Edc1–Dcp1–Dcp2 complex with either product m7GDP or capped RNA substrate have similar conformations in solution[19]. The most notable difference in the aligned substrate analog and product-bound Dcp2 structures is the difference in positioning of the phosphate chain of the cap. In the substrate analog-bound state, the β-phosphate, which undergoes nucleophilic attack by water during cap cleavage, is located 5.4 Å from the general base Glu152. This is very similar to the substrate-Glu distance in the prototypical Nudix hydrolase MutT, in which a solution-state NMR structure showed the catalytic Glu residue to be ~6 Å from the non-hydrolyzable AMPCPP substrate, with an intervening water molecule well-positioned for nucleophilic attack of the β-

phosphate[51]. In the post-catalytic, m7GDP product-bound structure, the β-phosphate of cap is positioned nearer to the Nudix helix, at a distance of 4.1 Å from the general base Glu152.

Substrate and product-bound structures also differ in the number of magnesium atoms located at the active site. In our substrate analog-bound structure, we were able to build one magnesium atom that is coordinated by Glu148. Although there is some additional density present in the $F_o–F_c$ difference map at the active site (Supplementary Fig. 6a), this was too weak to reliably build additional $Mg^{2+}$ or water. This agrees well with a high-resolution structure of the *S. cerevisiae* Dcp2 CD, which showed a single magnesium atom coordinated directly to *Sc* Glu149 (*Kl* Glu148), which is coordinated by other conserved Glu residues through water-mediated contacts. In both the *K. lactis* and *S. pombe* product-bound structures, however, three magnesium atoms are placed in the Dcp2 active site, coordinated by the conserved CD Glu residues and the phosphate groups of m7GDP (Supplementary Fig. 6b,c). Enzymology from our lab has shown that the $Mg^{2+}$-dependence of decapping kinetics fits a hill coefficient of 2.4, suggesting Dcp2 binds at least two metal ions[52], which is most consistent with the product-bound structures. In the end, both our substrate analog-bound structure (2.84 Å resolution) and the product-bound structures (3.1 Å and 3.5 Å for PDB 5N2V[37] and 5LOP[22], respectively) are not at sufficient resolution to very accurately distinguish $Mg^{2+}$ and water molecules, and high-resolution structures of Dcp2 with bound substrate and product will be required to unambiguously determine the location and precise role of coordinated magnesium and water molecules in the active site.

**Dcp2 shows selectivity for the first transcribed nucleotide.** As discussed above, our structure of substrate analog-bound Dcp2 suggests that the first transcribed nucleotide of RNA is recognized by the conserved aromatic Phe223 (*Sp* Tyr220). Surprisingly, although mutation of this residue produces temperature-sensitive growth defects in yeast[26], Tyr220Ala mutations in our single-turnover in vitro decapping assays with *S. pombe* Edc1–Dcp1–Dcp2 or Dcp1–Dcp2 complex had no effect on catalysis (Fig. 2a,b; Supplementary Fig. 4). In the apo, inactive conformation of Dcp1–Dcp2 that predominates in solution[19], Tyr220 (*Kl* Phe223) interacts with conserved aromatic W43 on the NRD of Dcp2 (e.g., PDB 2QKM[17], 5J3Y[20], 5KQ1[21]), which likely stabilizes the inactive conformation of the enzyme. Furthermore, we have recently shown that Tyr220 plays an important role in mediating Dcp2 conformational dynamics and that mutation of this residue alleviates an autoinhibited conformation of the apo enzyme[42]. It is therefore likely that Tyr220 is a multifunctional residue, which stabilizes an inactive/autoinhibited conformation of Dcp2 in the apo state and acts as a nucleotide binding platform in the substrate-bound, active conformation. In this way, mutation of Tyr220 may both accelerate catalysis by alleviating autoinhibition and impair catalysis by causing defects in substrate binding, resulting in decapping kinetics that appear relatively unaffected by Tyr220Ala mutation.

It was recently reported that Dcp2-mediated decapping in mammals is inhibited by m6A modification of the first transcribed nucleotide of RNA substrates[15], suggesting that Dcp2 may be selective for the identity of the first transcribed nucleotide, as is observed in the bacterial decapping enzyme NudC[53]. Accordingly, we next asked if Dcp2 selectivity for the first transcribed nucleotide of RNA substrate depends on Tyr220. We compared decapping kinetics for cap-radiolabeled 29-mer RNA substrates where the first transcribed nucleotide was either guanosine (G-RNA), adenosine (A-RNA), or cytosine (C-RNA) using WT, or Tyr220Gly *Sp* Edc1–Dcp1–Dcp2 decapping

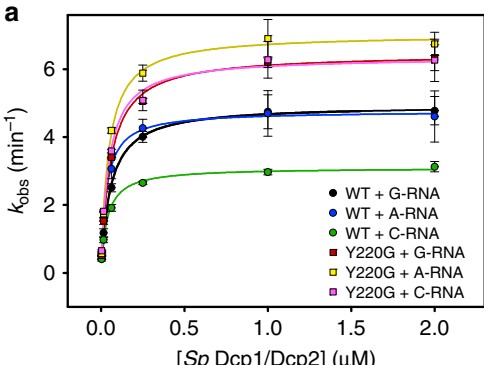

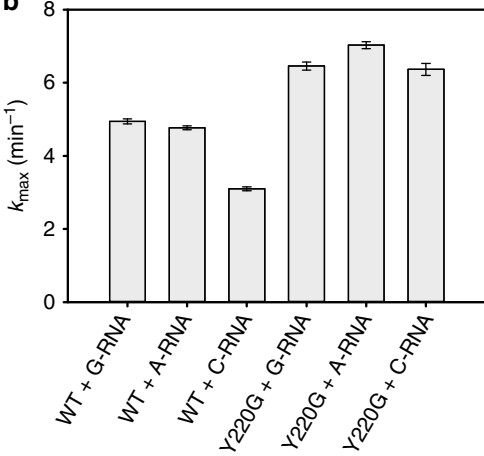

**Fig. 4** Dcp2 selectivity for the first transcribed nucleotide of RNA depends on Tyr220 (*Kl* Phe223). **a** Plots of $k_{obs}$ for decapping a 29-mer RNA substrate with G, A, or C as the first transcribed nucleotide (G-RNA, A-RNA, C-RNA, respectively), with WT or Tyr220Gly *S. pombe* Edc1–Dcp1–Dcp2 complex versus enzyme concentration. The concentration of the Dcp1–Dcp2(1–243) complex is varied, with Edc1(155–180) coactivator peptide kept at saturating concentration (50 μM). Errors on $k_{obs}$ are s.d. of two independent replicates. **b** Bar graph showing $k_{max}$ values determined from $k_{obs}$ versus enzyme concentration plots in (**a**). Errors are standard error of the fits to determine $k_{max}$

complexes (Fig. 4; Supplementary Table 2). Tyr220Gly mutations were used here to ensure maximal disruption of hydrophobic stacking interactions with substrate; this mutation also resulted in a small acceleration of decapping, likely resulting from disruption of the Y220–W43 interaction in the inactive Dcp2 conformation discussed above. With WT decapping complex, $k_{max}$ for both G-RNA and A-RNA is ~2-fold faster than C-RNA. Upon mutation of Tyr220, decapping is accelerated and the selectivity of 5′ G/A versus C nucleotides is abolished, resulting in approximately the same $k_{max}$ value for G-RNA, A-RNA, and C-RNA. This suggests that Dcp2 discriminates between G/A versus C at the first transcribed nucleotide position during the catalytic step of decapping, and that this selectivity critically depends on Tyr220 (*Kl* Phe223). Dcp2's Tyr220-dependent preference for G/A versus C at this position is in line with the observation that there is a strong genome-wide bias, in both fission yeast[54] and humans[55], toward purines (G,A) over pyrimidines (C,U) at the transcription start site +1 position. Furthermore, our selectivity data supports the hypothesis that Tyr220 (*Kl* Phe223) is important for recognition of the first transcribed nucleotide and that the apparent lack of kinetic defects, or even acceleration of decapping, when this residue is mutated may arise from its

multi-functional role in different conformations of the enzyme[19,42].

Finally, we also asked whether the activated *S. pombe* decapping complex showed selectivity for A versus m$^6$A at the first transcribed nucleotide position of RNA, as has been recently shown for mammalian Dcp2[15]. In contrast to the mammalian decapping enzyme, we found that fission yeast Edc1–Dcp1–Dcp2 complex, or Dcp2 alone, showed no selectivity for A versus m$^6$A at the 5′-most nucleotide position on a 29-mer RNA substrate in in vitro decapping assays (Supplementary Fig. 7). This suggests that recognition of the m$^6$A modification at the first transcribed nucleotide of mRNA (m$^6$A$_m$ in mammals, where this nucleotide is also 2′-O-methylated) by Dcp2 may only be conserved in higher eukaryotes.

## Discussion

Several recent studies have provided new structural information about the mRNA-decapping enzyme Dcp2[19–22]. These include structures of the Dcp1–Dcp2 heterodimer with markedly different conformations of Dcp2, bound cap analog or m$^7$GDP product, and in complex with Edc1-like or Edc3 coactivators (Fig. 5a). Here we expand on the structural understanding of Dcp2 activation and catalysis with a new structure of the four-protein complex Edc1–Dcp1–Dcp2–Edc3 in the catalytically active conformation of Dcp2 with a tight-binding substrate analog bound to the enzyme active site (Fig. 1a–c, Table 1). The coactivator Edc1 peptide engages conserved residues on Dcp1–Dcp2 NRD, providing a complementary surface against which the Dcp2 CD and flexible hinge pack in the catalytically active conformation, thus consolidating this conformation and promoting catalysis (Fig. 1d,e and 2) Cap substrate analog binds to the enzyme active site in an orientation that positions the RNA body to follow an extended, positively charged helix of Dcp2 to the bound Edc3 coactivator (Fig. 3), which has been shown to enhance RNA binding to these complexes in vitro[22]. Finally, we show that the conserved aromatic residue Tyr220 (*Kl* Phe223) that binds the second nucleotide of our substrate analog is critical for recognition of the first transcribed nucleotide in RNA by the decapping complex (Fig. 4). Purine specificity is conferred for this nucleotide during the catalytic step of decapping.

Many different conformations of the decapping enzyme Dcp2 have been structurally characterized, but a consensus is now emerging around the identity of the catalytically active conformation of Dcp2. With the addition of the Edc1–Dcp1–Dcp2–Edc3 cap analog structure presented here, we now have both product m$^7$GDP and substrate analog-bound structures of the Dcp1–Dcp2 decapping complex in what can safely be described as the active conformation (Fig.5a, active structures). The substrate and product-bound active conformations are nearly identical, with major differences only in the positioning of the cap phosphate chain before and after hydrolysis (Fig. 3d). These structures all show that the m$^7$G nucleotide of cap is specifically recognized by stacking with a conserved Trp and hydrogen bonding to a conserved Asp residue on the Dcp2 NRD, explaining why the NRD contributes 100-fold to catalysis[31]. In the active conformation, the 190s loop of the Dcp2 CD packs against the opposite face of the m$^7$G base, which brings the Nudix helix and catalytic Glu within ~6 Å of the cap β-phosphate, close enough to carry out hydrolysis chemistry and cap cleavage (Fig. 3b).

The catalytic importance and roles of the additional conformations of Dcp2 that have been structurally characterized, but are very different from the active conformation (Fig. 5a, inactive, RNA-binding, and pre-catalytic structures; Fig. 5b,c), remain less clear. In our previously published pre-catalytic substrate analog-bound PNRC2–Dcp1–Dcp2 structure[21], Dcp2 adopts a conformation in which the NRD recognizes one m$^7$G nucleotide of cap analog as in the active conformation, but the CD of Dcp2 is rotated 90° and *Sp* Tyr220 stacks against the opposite face of this m$^7$G, rather than the 190s loop (Fig. 5a, pre-catalytic; Fig. 5c). In the pre-catalytic conformation, the second m$^7$G nucleotide of our two-headed cap analog is bound by the strictly conserved *Sp* Tyr92 residue near the Dcp2 hinge. In the Edc1-bound active conformation, however, this Tyr is occluded by interaction with the Dcp2-activating motif on Edc1 (Fig. 5d). It is possible that this pre-catalytic conformation, in which cap analog is bound by a composite nucleotide binding site consisting of conserved residues on the Dcp2 CD and NRD, represents an early step in cap recognition that precedes formation of the active conformation and its consolidation by the Edc1 YAGxxF motif. Likewise, the apo Edc1–Dcp1–Dcp2 structure by Valkov et al.[20], which adopts a conformation that positions RNA-binding residues on the Dcp2 CD and Dcp1 opposite each other to create a positively charged channel, may represent a conformation of the apo enzyme that is primed for RNA binding or scanning[21,24] (Fig. 5a; RNA-binding). With both of these alternative conformations, however, it is so far difficult to rule out the possibility that these are simply off-pathway conformations captured by crystallization. This is not to say that valuable information is not provided by conserved cap or coactivator contacts with the enzyme, but rather that these specific conformations may not be on-pathway in the Dcp2 catalytic cycle. Indeed, recent PRE NMR data suggests that the apo Edc1-bound conformation in the Valkov et al. structure is not populated in solution in vitro[19]. In these same experiments, it was not possible to distinguish our pre-catalytic substrate-bound conformation from the closed, inactive conformation of Dcp2 that predominates in solution (Fig. 5a, inactive; Fig. 5b).

Building on recent reports of Dcp2 autoinhibition in yeast[29], our lab has now shown that the C-terminal extension in fission yeast Dcp2 contains autoinhibitory motifs that interact with the core domains of Dcp1–Dcp2 to inhibit the decapping complex and impair activation by Edc1, and that this autoinhibition is alleviated by coactivator Edc3[42]. The Edc1–Dcp1–Dcp2–Edc3 structure presented here provides the structural framework to begin to understand how coactivators act to alleviate autoinhibition and activate catalysis. We suggest a model of decapping regulation whereby full-length Dcp2 is autoinhibited and coactivators Edc3 and Edc1 act simultaneously to (1) alleviate this autoinhibition, and (2) activate decapping catalysis by both consolidating the Dcp2 active conformation and by enhancing RNA binding to the complex. In this way, switch-like mutli-log unit changes in decapping activity are easily achievable when multiple coactivators bind Dcp2 and function simultaneously. Because many decapping coactivators have genetic or physical interactions with one another[29,45–49], we anticipate that combinatorial control of decapping via multiple coactivators is a general and important principle for regulation of mRNA decay.

## Methods

**Protein expression and purification.** *K. lactis* his-tev-Dcp1(1–188)–Dcp2(1–275, E152Q)-Edc3(1–66) polycistronic construct used for crystallization experiments was obtained as a single *E. coli* codon-optimized DNA sequence with a T7 promoter preceding each protein as a gBlock from Integrated DNA Technologies (his is hexahistidine affinity tag, tev is Tobacco Etch Virus protease cleavage site; see Supplementary Table 3 for sequence). This was cloned into a Kanamycin-resistant expression vector and transformed into *E. coli* BL21-star DE3 cells (Invitrogen; see Supplementary Table 4 for primers used). Cells were grown in LB media to OD ~0.6, protein expression was induced with IPTG for 18 h at 20 °C. Cells were harvested at 5000 × *g*, lysed by sonication, and clarified at 16,000 × *g* in lysis buffer (50 mM sodium phosphate pH 7.5, 300 mM sodium chloride, 10 mM imidazole, 5% glycerol, 10 mM 2-mercaptoethanol, Roche EDTA-free protease inhibitor cocktail). The protein complex was purified by Ni-NTA affinity chromatography

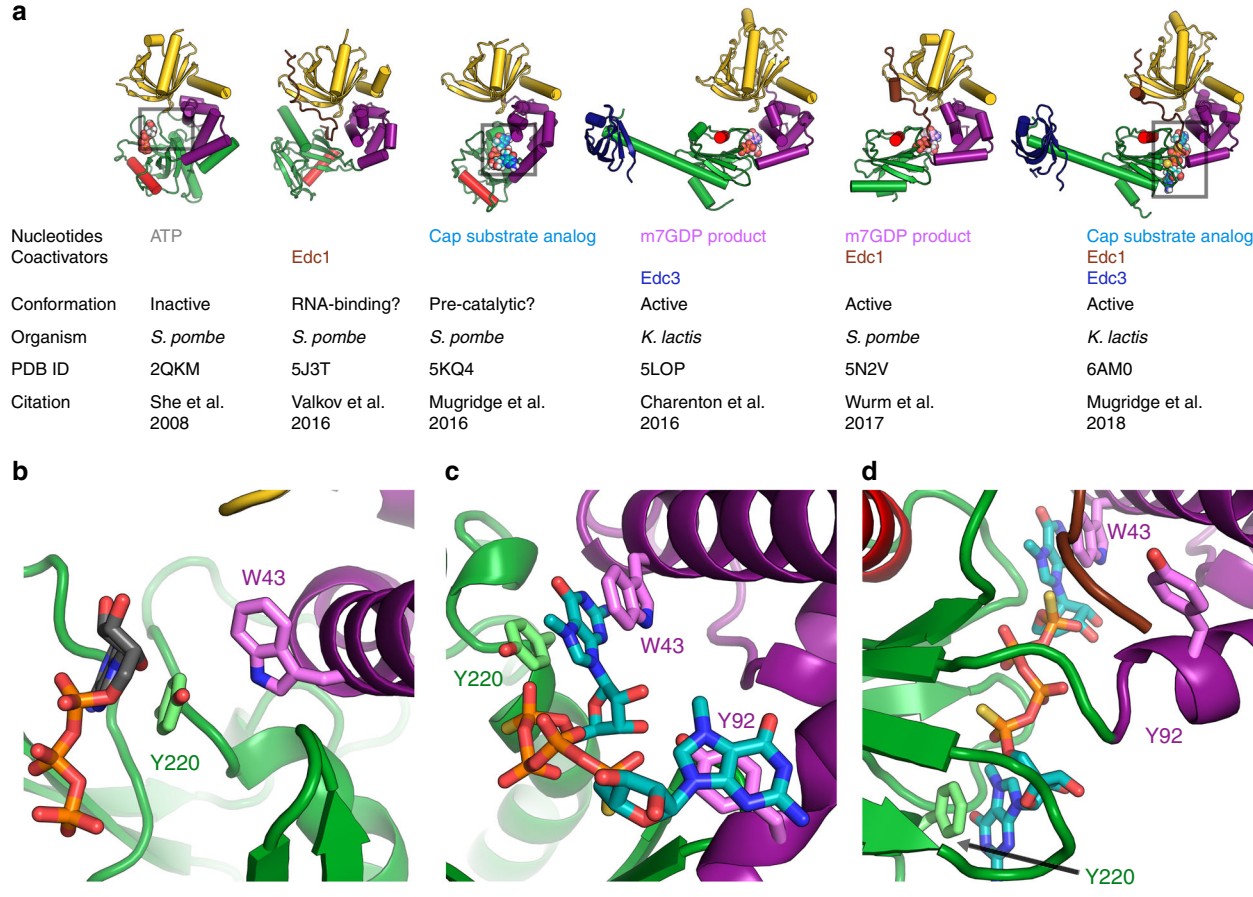

**Fig. 5** Summary of recent Dcp1–Dcp2 structures and interactions between important aromatic cap-binding residues in different conformations. **a** Gallery showing the different overall conformations of recent Dcp1–Dcp2 structures. Text lists bound nucleotides, coactivators, and proposed description of the Dcp2 conformation. Dcp1 is yellow, Dcp2 NRD is purple, Dcp2 CD is green with the catalytic Nudix helix colored red, Edc1 is brown, Edc3 is dark blue, cap substrate analog is cyan, m$^7$GDP product is pink. From left to right, PDB codes and citations are 2QKM[17], 5J3T[20], 5KQ4[21], 5LOP[22], 5N2V[19], 6AM0 (this work). **b** Close-up view of ATP-bound inactive structure (PDB 2QKM)[17]. Conserved, cap-binding residues Trp43 and Tyr220 make edge-on contacts in this inactive conformation. ATP is gray. **c** Close-up view of cap analog-bound, pre-catalytic structure (PDB 5KQ4)[21]. Residues Tyr220 and Trp43 bind one m$^7$G nucleotide of two-headed cap analog, Tyr92 binds the second m$^7$G nucleotide of the cap analog. **d** Close-up view of cap analog-bound, active structure (PDB 6AM0); *S. pombe* residue numbering is used here instead of *K. lactis* for comparison with structures shown in (**b**) and (**c**). Trp43 (*Kl* Trp49) binds one m$^7$G nucleotide of two-headed cap analog and Tyr220 (*Kl* Phe223) binds the second m$^7$G of cap analog, mimicking binding of the first transcribed nucleotide. Tyr92 (*Kl* Tyr98) is buried in a protein–protein interface with Edc1 coactivator in the active conformation

and solubility/affinity tags were cleaved by treatment with TEV protease overnight at room temperature in elution buffer (50 mM sodium phosphate pH 7.5, 300 mM sodium chloride, 250 mM imidazole, 10 mM 2-mercaptoethanol). The *Kl* Dcp1–Dcp2–Edc3 complex was separated from cleaved tags by size exclusion chromatography on a GE Superdex 200 16/60 gel filtration column in crystallization buffer (10 mM HEPES pH 7, 150 mM NaCl, 1 mM DTT). The purified complex was concentrated to 15 mg/mL and flash frozen in liquid nitrogen for crystallization experiments.

*S. pombe* his-GB1-tev–Dcp1(1–127)–Dcp2(1–243) constructs used for kinetic experiments were obtained by cloning the genomic *Sp* Dcp2(1–243) coding sequence into MCS2 of Novagen pACYC co-expression vector, and an *E. Coli* codon-optimized *Sp* Dcp1(1–127) DNA sequence obtained from Integrated DNA Technologies into MCS1 of the pACYC vector containing an N-terminal his-GB1-tev tag (GB1 is a solubility tag derived from the B1 domain from Protein G; see Supplementary Table 3 for codon-optimized *Sp* Dcp1 sequence). The *Sp* Dcp1–Dcp2 protein complex was expressed and purified as described above for the *Kl* Dcp1–Dcp2–Edc3 complex. After TEV cleavage, the complex was separated from cleaved tags by size exclusion chromatography on a GE Superdex 200 16/60 gel filtration column in storage buffer (50 mM HEPES pH 7.5, 100 mM NaCl, 5 mM DTT). The purified *Sp* Dcp1–Dcp2 complex was concentrated, glycerol added to a final concentration of 20%, and flash frozen in liquid nitrogen; final concentration was 200 µM. Point mutants of the *Sp* Dcp1–Dcp2 complex were made by whole plasmid PCR with mutagenic divergent primers, verified by

sequencing, and purified as above (see Supplementary Table 4 for primer sequences).

**Crystallization of Edc1–Dcp1–Dcp2–Edc3 with substrate analog.** *Kl* Edc1(355-380) peptide (from gene KLLA0_A01474g) was chemically synthesized by Peptide 2.0 and dissolved in crystallization buffer to a concentration of 5 mM. Two-headed cap substrate analog, m$^7$Gp(S)ppp(S)m$^7$G[50,56], was dissolved in water at a concentration of 100 mM and the pH was adjusted to ~7.

Protein solution consisting of 4 mg/mL purified *K. lactis* Dcp1(1–188)–Dcp2 (1–275, E152Q)–Edc3(1–66), 1 mM *Kl* Edc1 peptide, 6 mM substrate analog, and 10 mM MgCl$_2$ was prepared in crystallization buffer and incubated at room temperature for 30 min. Using a TTP Labtech Mosquito Crystal robot, 250 nL of protein solution was combined with 250 nL of well solution, and then 50 nL of seed stock, in a hanging-drop vapor diffusion experiment. Crystallization drops were prepared at room temperature and immediately moved to 4 °C. The well solution contained 220 mM magnesium acetate, 30 mM EDTA, and 8% w/v PEG 8000. The seed stock was prepared using Hampton's Seed Bead kit, using crystals grown with 1:1 protein solution and well solution containing 200 mM magnesium acetate and 10% w/v PEG 8000 at room temperature, diluted 10,000-fold. Block-shaped crystals ~40–80 µm in size grew at 4 °C within 1–2 days. Crystals were flash frozen in liquid nitrogen using a cryoprotectant consisting of well solution with 15% PEG 8000.

**Crystallographic data collection and refinement**. All data were collected at Lawrence Berkeley National Lab, Beamline 8.3.1 at the Advanced Light Source on a Dectris Pilatus3 S 6 M detector at 100 K and a wavelength of 1.11583 Å. Data were indexed, integrated and scaled using XDS[57]. The structures were solved in space group C2 by molecular replacement using Phaser in the CCP4 suite[58]. Fragments of chains A, B, and C of PDB 5LOP were used as molecular replacement models[22], first placing Dcp1–Dcp2 NRD(1–96), then Dcp2 CD(104–240), then Dcp2 (252–260)–Edc3(1–66). The asymmetric unit contained two nearly symmetric copies of the Edc1–Dcp1–Dcp2–Edc3 complex. In the crystal lattice, one molecule of Edc3 Lsm domain forms a dimer as in the Kl Edc3–Dcp1–Dcp2–m7GDP product structure (PDB 5LOP)[22], but the spacegroups (C2 for AM60 and P6₄22 for 5LOP) and overall crystal packing are entirely different between our structure and the previously determined K. lactis structure. Molecular replacement failed to correctly place the Dcp2(252–260)–Edc3(1–66) fragment for chain H, as the density was very weak for this chain of Edc3, likely because this molecule of Edc3 does not form a dimer in the crystal lattice. This fragment was placed manually in COOT, based on strong density for the Dcp2(252–260) Edc3-binding motif and strong density for some β-strands in Edc3. After a preliminary round of refinement including rigid-body refinement in PHENIX[59], the Edc1 peptide, Dcp2 hinge, and Dcp2 extended helix were built manually in COOT[60]. The structure was then iteratively refined in PHENIX and manually adjusted in COOT. Final refinement used Non-Crystallographic Symmetry, Translation-Libration-Screw parameterization for chain H, and secondary structure, stereochemical and atomic displacement parameter restraints. For the final model of Kl Edc1–Dcp1–Dcp2–Edc3, all residues were in favored (95.6%) or allowed (4.4%) regions of the Ramachandran plot as evaluated by MolProbity[61].

**Structural analysis and visualization**. All alignments and structural figures were prepared using PyMol version 1.7. Electrostatic surfaces were calculated using PDB2PQR & APBS webservers (http://nbcr-222.ucsd.edu/pdb2pqr_2.0.0/)[62,63], using the AMBER94 forcefield to calculate charges. Electrostatic surfaces were visualized in PyMol using APBS tools 2.1. Buried surface area in protein interfaces was calculated using the PDBePISA webserver (http://www.ebi.ac.uk/pdbe/prot_int/pistart.html)[64].

**Kinetic decapping assays**. For the decapping assays measuring $k_{obs}$ for Dcp1–Dcp2 point mutants ±Edc1 described in Fig. 2 and Supplementary Fig. 4, Sp Dcp1–Dcp2(1–243) WT and mutant complexes had a final concentration of 2 μM and Sp Edc1 (155–180) peptide had a final concentration of 100 μM. Sp Edc1 (155–180) peptide was chemically synthesized by Peptide 2.0 and stock solutions were prepared at 5 mM concentration in crystallization buffer, flash frozen in liquid nitrogen, and stored at −20 °C. These experiments used an in vitro transcribed, enzymatically capped 355-mer RNA substrate derived from MFA2 and ³²P-labeled at the α-phosphate of cap. Reactions were carried out in the decapping buffer used previously to measure PNRC2 activation kinetics (50 mM HEPES pH 7, 100 mM KCl, 10% glycerol, 5 mM MgCl₂, with RNase inhibitor and 0.1 mg/mL acetylated BSA)[21]. 3× protein solutions with Dcp1–Dcp2+Edc1 peptide, and 1.5× capped RNA solutions were equilibrated separately at 4 °C for 30 min prior to initiation. Reactions were initiated by combining 15 μL 3× protein solution with 30 μL 1.5× RNA solution at 4 °C. Time points were quenched by the addition of excess EDTA, TLC was used to separate RNA from product m7GDP, and the decapped fraction was quantified using a GE Typhoon scanner and ImageQuant software. Plots of fraction m7GDP product versus time were fit to a first order exponential to obtain $k_{obs}$; in the case of R33A and I96G point mutants, decapping kinetics were too slow to obtain reliable exponential fits and $k_{obs}$ was obtained from a linear fit of initial rates (less than 20% decapped), by dividing the slope by the average endpoint of 0.95. Errors on individual time points in Fig. 2a are the s.d. between two independent repetitions of the decapping reaction; errors on $k_{obs}$ in Fig. 2b are s.d. of $k_{obs}$ values obtained from these two independent replicates.

For decapping assays measuring kinetics with G versus C at the first transcribed nucleotide described in Fig. 4 and those measuring $K_M$ and $k_{max}$ for Edc1 activation (Supplementary Fig. 5, Supplementary Table 1), synthetic 5′-triphosphate 29-mer RNA oligonucleotides with either guanosine (G-RNA) or cytosine (C-RNA) as the 5′ nucleotide were obtained from Trilink BioTechnologies and enzymatically capped with ³²P-radiolabeled GTP, as above. In the decapping reaction, Sp Dcp1–Dcp2(1–243) had a final concentration ranging from 2 μM to 4 nM, the final concentration of Sp Edc1(155–180) peptide was held constant at 50 μM, and capped RNA concentration was <100 pM. For these experiments, glycerol was removed from the decapping buffer described above in order to obtain more easily saturable kinetics with the 29-mer RNA substrate. Decapping reactions were prepared, executed, and analyzed as described above. To obtain $k_{max}$ and $K_M$, $k_{obs}$ was plotted versus protein concentration and fit to the model: $k_{obs} = k_{max}[E]/K_M + [E]$. Errors on $k_{obs}$ values shown in Fig. 4a are s.d. of two independent replicates. Errors on $k_{max}$ values shown in Fig. 4b are the standard error from fits to determine $k_{max}$ and $K_M$.

**Data availability**. Coordinates and structure factors were deposited in the Protein Data Bank with accession code PDB 6AM0. Other data are available from the corresponding author upon reasonable request.

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

## Acknowledgements

We thank J. Holton and G. Meigs at Lawrence Berkeley National Laboratory, Advanced Light Source beamline 8.3.1, and X. Liu for help with X-ray data collection and refinement. We also thank J. Kowalska for helpful discussions on the design and synthesis of the two-headed cap analog. This work was supported by the US National Institutes of Health (R01 GM078360 to J.D.G. and NRSA fellowship F32 GM105313 to J.S.M.), The Foundation for Polish Science (TEAM/2016-2/13 to J.J.), the National Science Centre, Poland (fellowship no. UMO-2014/12/T/NZ1/00528 to M.Z.), the Genentech Foundation Fellowship (to R.W.T), and the UCSF Discovery Fellowship (to R.W.T). The Advanced Light Source is supported by the US Department of Energy under contract no. DE-AC02-05CH11231.

## Author contributions

J.S.M. designed and purified all protein constructs, carried out crystallization experiments, collected and refined crystallographic data, wrote the manuscript, and prepared the figures. J.S.M. and R.W.T. carried out decapping kinetics experiments. M.Z. and J.J. designed and synthesized the two-headed cap analog. J.D.G. supervised the project and experimental design and guided manuscript preparation and editing. All authors read and commented on the manuscript.

## Additional information

**Competing interests:** The authors declare no competing interests.

