## [Peer Review File(PDF 274 kb) · Nature Communications]

Reviewers' comments:

Reviewer #1 (Remarks to the Author):

The manuscript "Structure of the activated Edc1-Dcp1-Dcp2-Edc3 mRNA decapping complex with substrate analog poised for catalysis" by Mugridge et al. presents the X-ray crystal structure of *K. lactis* Dcp1-Dcp2 in complex with coactivators Edc1 and Edc3 and with a 2-headed cap substrate analog. This structure represents the "active" conformation of Dcp2, achieved upon binding of the cap analog as well as Dcp1 and the coactivators. The orientation of the substrate analog in the active site of Dcp2 allows one to model the path of the RNA from the active site, along a positively charged helix of Dcp2 to the coactivator Edc3, which is thought to enhance decapping by increasing RNA binding efficiency. The authors also suggest an explanation for preference for a purine as the first transcribed nucleotide in RNA. The models generated from structural studies have been tested by *in vitro* decapping assays, where the roles of residues in Dcp2 NRD and CD in the process of activation and nucleotide recognition have been tested.

The structural work presented in this study is sound. However, given that the authors have a fairly low R_{free} for a structure at 2.8Å, it would be helpful to mention the search model used for molecular replacement in the main text (although the entire process of structure determination is very well described in the Methods section).

Two major concerns are outlined below:

1) The proposed model for selectivity of the first transcribed nucleotide of RNA does not appear to be very robust. Mutation of Tyr220 to alanine does not, understandably, lead to a defect in decapping *in vitro*. However, mutation of Tyr220 to glycine leads to an increase in rate of decapping. The reason for this is not very clear, although the authors allude to this residue possibly playing a role in maintaining the auto-inhibited state of Dcp2. In that case, why does mutation to alanine not lead to an increase in decapping rates? Furthermore, the difference in *k*_{max} between a purine and a pyrimidine residue (G and C, respectively) is a modest two-fold. It is unclear if this two-fold difference really translates to a clear preference for a purine nucleotide in cells. It would be worthwhile for the authors to test the difference between an A-RNA (where A is the first transcribed nucleotide) and other RNAs, since a study carried out long ago (Wei C et al., *Nature* 1975) suggest that about 30% of capped mRNAs contains a N₆, 2'O dimethylated adenine.

2) The structure presented here is the first one of Dcp1-Dcp2 bound to two coactivators, Edc1 and Edc3. However, structures of Edc1 and Edc3 individually bound to Dcp1-Dcp2 are already available. As such, this structure appears to be a sum of its two subparts (Edc1-Dcp1-Dcp2 and Edc3-Dcp1-Dcp2), which is not unexpected. Furthermore, the authors have previously reported the structure of a complex of Dcp1-Dcp2 with PNRC2, another coactivator. Also structures of Dcp2 in different stages of its activity (inactive, RNA-binding, pre-catalytic, active) are known. In light of all the structural (X-ray crystallographic and NMR) data as well as kinetic studies available, the new insights presented by the structure of Edc1-Dcp1-Dcp2-Edc3 are not sufficiently clear and not very well highlighted in the manuscript. Unfortunately, one cannot derive information about the catalytic mechanism from this structure, as it is simply not at high enough resolution to place water and magnesium ions unambiguously. Indeed, the authors only see one magnesium ion of the two that are necessary for catalysis. Therefore, it might be more interesting and insightful if the authors were to focus on the "path of RNA binding" from the active site to the Edc3 binding site of Dcp2. Mutational analysis carried out *in vitro*, with a longer RNA substrate, or in yeast might provide more support for this model and reveal information on substrate channeling to Dcp2, which would be of great interest to the reader.

Minor points:

- 1) Figure 1d and line 173 of text: ..."Phe364 plugs into a positively-charged/aromatic pocket at the Dcp1-Dcp2NRD interface". It would be great if the authors could add some of the positively charged/aromatic residues that make up this interface in Figure 1d.
- 2) Line 269 of text – Fig. 4d should be Fig. 3d (there is no Figure 4d).
- 3) One should ideally report mean values/s.d from at least 3 independent experiments, even if the errors appear to be very low.

Reviewer #2 (Remarks to the Author):

The manuscript by Mudridge, et al entitled "Structure of the activated Edc1-Dcp1-Dcp2-Edc3 mRNA decapping complex with substrate analog poised for catalysis" presents the first crystal structure of the Dcp1-Dcp2 decapping machinery bound to coactivators Edc1 and Edc3 in the presence of a substrate analog. The structural work is supported by kinetic analysis of Dcp2 mutants to show that Dcp2 is selective for the first transcribed nucleotide in the RNA substrate.

This work is significant because it helps clarify the active conformation of the decapping complex. Several Dcp1-Dcp2 structures have been published over recent years, revealing multiple conformational states that appear to be dependent on the presence of various substrates and co-factors. In this new substrate-bound structure, Dcp2 adopts the same conformation as previously published product-bound structures.

The manuscript is very well written. The structure appears to be well-refined and I have no substantive concerns with the data or the presentation of the data. The summary/overview of existing structures in Figure 5 is particularly helpful. For clarity, I recommend adding the organism (since the structures being compared don't all originate from the same organism) and the PDB ID to Figure 5a (rather than putting it solely in the figure legend).

Minor corrections:

Ln 269 – Fig. 4d should be Fig 3d.

Ln 386 – place a comma between "conformations" and "however"

Point-by-point response to reviewer comments for manuscript NCOMMS-17-21908-T

Structure of the activated Edc1-Dcp1-Dcp2-Edc3 mRNA decapping complex with substrate analog poised for catalysis

by Jeffrey S Mugridge, Ryan W Tibble, Marcin Ziemniak, Jacek Jemielity & John D Gross

Overall, we were very pleased to receive generally positive reviews for our manuscript “Structure of the activated Edc1-Dcp1-Dcp2-Edc3 mRNA decapping complex with substrate analog poised for catalysis”. Reviewer 1 considered the structural work “sound” and well-described, and Reviewer 2 was very positive, noting that “this work is significant because it helps clarify the active conformation of the decapping complex”, that the manuscript was “well-written”, and that they had “no substantive concerns with the data or the presentation of the data”.

Reviewer 1 raised two major questions and asked for some new data to be added, and Reviewer 2 suggested only minor changes. Additionally, the editors raised one concern about an unpublished manuscript from our lab, which we cite at different points in our work.

In response to these comments, we have added new data requested by Reviewer 1, and we believe addressed all of the questions, comments, and concerns raised by reviewers with changes to the manuscript. We thank both reviewers for their careful consideration of our work, and believe the revised manuscript is stronger after addressing and incorporating their helpful comments. Below is our point-by-point response to these questions, which outlines the significant changes made to the revised manuscript.

Re: Reviewer 1 comments

Reviewer 1 raised two major concerns about the initial manuscript:

1. Reviewer 1 asked about the robustness of selectivity for the first transcribed nucleotide of RNA, as we observed only ~2-fold effects for different first transcribed nucleotides, and suggested we test decapping for A-RNA (A as the first transcribed nucleotide) because many mRNAs are known to contain the m⁶A_m modification at the 5' nucleotide.

We prepared capped RNA substrates with A or m⁶A (A-RNA and m⁶A-RNA, respectively) at the 5'-most nucleotide position and characterized their kinetic parameters in single turnover decapping assays. We find that A-RNA is decapped by WT decapping complex with the same k_{\max} as G-RNA, which is ~two-fold faster than C-RNA. G-RNA, A-RNA, and C-RNA are all decapped with nearly identical k_{\max} by Y220G decapping complex. This new data has been incorporated into Figure 4, Supplementary Table 2, and text on page 11; it is consistent with and adds further support to our conclusions that Dcp2 has a weak preference for purines at the first transcribed nucleotide, in line with the genome-wide preference for purines at the transcription start site +1 position. While we of course cannot say (and do not say) whether this modest *in vitro* preference translates to genome-wide differences in decapping in cells, it does support our structural data showing that Y220 recognizes the first transcribed nucleotide of the RNA substrate.

We also found that A-RNA and m⁶A-RNA were decapped with identical rates by fission yeast WT decapping complex. This is in contrast to a recent report that showed that human Dcp2 decaps m⁶A-RNA slower than A-RNA (Mauer et al 2017). We have added this data as Supplementary Fig. 7 and a description of these experiments to the main text on page 11. We suggest that the recognition of m⁶A at the first transcribed nucleotide of mRNA may only be conserved in higher eukaryotes; indeed, 2'-O-methylation of the first transcribed nucleotide is a modification found only in higher eukaryotes and not in yeast.

As part of concern 1, Reviewer 1 also asked why there was a difference in the decapping rates between Y220A and Y220G mutations.

In the apo, inactive conformation of Dcp2 (characterized structurally in She et al 2008, and shown to be the dominant solution conformation in Wurm 2017), Y220 on the CD of Dcp2 makes a hydrophobic interaction with W43 on the NRD, and this interaction likely stabilizes the inactive conformation of Dcp2. Therefore, mutation of Y220 can disrupt this interaction and result in activation of the enzyme. Gly mutation is more activating than Ala mutation simply because Gly more completely disrupts the hydrophobic 220-W43 interaction to a greater extent than the more hydrophobic Ala.

We have added a clause to page 11 to help make this more clear: "this mutation also resulted in a small acceleration of decapping, likely resulting from disruption of the Y220-W43 interaction in the inactive Dcp2 conformation discussed above"

2. The second related set of points raised by reviewer 1 suggests that "this structure appears to be a sum of its two subparts (Edc1-Dcp1-Dcp2 and Edc3-Dcp1-Dcp2), which is not unexpected", that "the new insights presented by the structure of Edc1-Dcp1-Dcp2-Edc3 are not sufficiently clear and not very well highlighted in the manuscript", and that therefore "it might be more interesting and insightful if the authors were to focus on the path of RNA binding from the active site to the Edc3" coactivator. Reviewer 1 suggests in vitro mutational analysis to proposed RNA binding residues to add further support to the RNA binding model, which they think "would be of great interest to the reader".

First, we agree with Reviewer 1 that highlighting the RNA binding model suggested by substrate positioning in our structure more clearly in this manuscript would be of interest to many readers. As was perhaps not very clear in our original manuscript, basic residues on the dorsal surface of Dcp2 have been extensively mutagenized, and their

effects on RNA binding, decapping activity, and cellular fitness, have been previously evaluated. Thus much of the analysis suggested by Reviewer 1 has already been carried out, and all of this biochemical and biophysical data agrees with the RNA binding model predicted by substrate positioning in the active site of our structure, where RNA extends down the positively-charged dorsal surface, away from the Dcp2 composite active site, toward coactivator Edc3.

In the revised manuscript, we have separated the RNA binding model into its own Results subheading, and carefully laid out all of the evidence and mutagenesis data supporting the RNA binding path suggested by our structure (page 8).

Second, to the reviewer's broader point that the significance of our Edc1-Dcp1-Dcp2-Edc3 is not sufficiently clear given other known Dcp2 conformations and structures with either Edc1 or Edc3 bound individually, we would note the following: (1) all previous structures with Dcp2 in the putative "active" conformation, including those with either Edc1 or Edc3 bound (Wurm et al 2017, Charenton et al 2016, respectively), were post-catalytic, product-bound structures. (2) We previously solved a structure with substrate-analog bound to Dcp2 in a very different conformation (Mugridge et al 2016), and there is an apo Edc1-Dcp1-Dcp2 structure in which Dcp2 adopts yet another conformation that the authors claimed was the "activated" conformation of Dcp2 (Valkov et al 2016). (3) The recent identification of these very different Dcp2 conformational states has produced some disagreement in the literature about their role in catalysis, the mechanism of Edc1 decapping activation, and the true identity of the catalytically-active conformation of Dcp2 (see reviews Collier 2016, Valkov et al 2017).

Given points (1) – (3) above, our structure is significant because it is the first structure to show how substrate is positioned in the Dcp2 active site, and thus clarifies the catalytically-active conformation of the enzyme (Reviewer 2 specifically mentions this point of significance). As Reviewer 1 notes, and we discuss above, substrate positioning in the active site also provides good evidence for an RNA binding path on Dcp2. Finally, the co-occupancy of Edc1 and Edc3 in the substrate-bound, active conformation of Dcp1/Dcp2 shows that these coactivators engage the substrate and product-bound complexes similarly (not a foregone conclusion), provides further support for the proposed modes of action for these coactivators, and highlights the idea that these coactivators can work synergistically to promote decapping by distinct mechanisms, as evidenced by genetic data (Refs Decourty et al 2008, He & Jacobsen 2015), and our upcoming report on autoinhibition (Paquette et al).

Reviewer 1 also made the following minor points:

Authors may add some of the residues that make up the Phe364 – Dcp1/Dcp2 interface mentioned on line 173 to Fig. 1.

We found that adding these residues to Figure 1 made panel d too complicated. So instead we added Supplementary Figure 3, which shows the Phe364 – Dcp1/Dcp2 NRD interface and the conserved residues that make contacts with Phe364. This is discussed briefly in the main text.

Change Fig. 4d to 3d. We have corrected this typo.

"One should ideally report mean values/s.d. from at least 3 independent experiments". While we certainly agree that more replicates are generally preferable, given the limited lifetime of the radioactive RNA substrates and the very high reproducibility of all of the kinetic experiments (across different RNA / protein preparations), we find that two independent replicates are a good compromise of efficiency and redundancy.

Given the "fairly low Rfree for a structure at 2.8Å, it would be helpful to mention the search model used for molecular replacement in the main text (although the entire process of structure determination is very well described in the Methods section)". We have added this to page 3 of the main text.

Re: Reviewer 2 comments

We were very happy to see that reviewer 2 found the results in our manuscript significant, well-written, and had no substantive concerns prior to publication.

Reviewer 2 suggested the following minor changes:

Add organism and PDB ID to Figure 5. We have added these to updated Figure 5.

Change Fig. 4d to 3d. We have corrected this typo.

Comma between conformations and however, line 386. OK, added.

REVIEWERS' COMMENTS:

Reviewer #1 (Remarks to the Author):

The authors have adequately addressed all my questions and concerns. I have no further questions; just one minor comment that I overlooked in the initial submission - I suggest that the authors add the details of the Ramachandran plot (# of residues favored, # allowed and # of outliers) to Table 1.